# Participation in Household Physical Activity Lowers Mortality Risk in Chinese Women and Men

**DOI:** 10.3390/ijerph20020987

**Published:** 2023-01-05

**Authors:** Lan Hu, Lu Wang, Yunquan Zhang, Ke Wang, Yaqi Wang, Huiyue Tan, Yin Zhang

**Affiliations:** 1Department of Nursing, Medical College, Wuhan University of Science and Technology, Wuhan 430065, China; 2Hubei Cancer Hospital, Tongji Medical College, Huazhong University of Science and Technology, Wuhan 430079, China; 3Department of Nursing, Wuhan 1st Hospital, Wuhan 430022, China; 4Department of Epidemiology and Biostatistics, School of Public Health, Wuhan University of Science and Technology, Wuhan 430065, China; 5Hubei Province Key Laboratory of Occupational Hazard Identification and Control, Wuhan University of Science and Technology, Wuhan 430065, China

**Keywords:** household physical activity, sport and physical exercise, mortality, dose–response

## Abstract

The health benefits of sport and physical exercise (SPE) have been well documented, while the influence of household physical activity (HPA) on health has received much less research attention. This study aims to provide epidemiologic insight into the role HPA plays in the development of all-cause, cardiovascular disease (CVD), respiratory disease (RESP), and cancer mortality in a nationwide cohort of Chinese adults. We conceived a prospective cohort comprising 30,791 participants aged ≥16 years from 25 provinces of China using data derived from baseline (2010) and 4 waves of follow-up (2012–2018) investigations of the China Family Panel Studies. Self-reported times of HPA and SPE were collected by interviewing participants with a standard questionnaire. Cox proportional hazard models were used to assess the associations of HPA and SPE with all-cause, CVD, RESP, and cancer mortality, adjusting for demographic and socioeconomic factors, lifestyle behaviors, and health status. A restricted cubic spline smoother was used to investigate the dose–response relationships of HPA and SPE with mortality outcomes. Sex subgroup analyses were conducted to examine the potential effect disparity between men and women. To investigate the interactive effects of HPA and SPE, we calculated the relative excess risk due to the interaction and attributable proportion of additive effects to the total observed effects. During a median follow-up of 7.2 years, a total of 1,649 deaths occurred, with 209 cases from CVD, 123 from RESP, and 323 from cancer. HPA was identified to be associated with reduced mortality outcomes, suggesting remarkably reduced risks of 43–60% in all-cause mortality, 42–50% in CVD mortality, 36–71% in RESP mortality, and 38–46% in cancer mortality. In general, higher levels of HPA tended to be associated with lower risks. An approximately inverted J-shape association was identified between HPA and all-cause and cause-specific mortality, suggesting strong evidence for potential nonlinearity. Women performing HPA had a lower risk of all-cause, CVD, and cancer mortality. We did not identify significant evidence for additive interaction between HPA and SPE. HPA is independently associated with a reduced risk of mortality in Chinese women and men. More biological studies are needed to validate our findings and clarify the mechanisms underlying the association.

## 1. Introduction

The potential health benefits of physical activity (PA) have been well documented in numerous epidemiologic studies [1,2,3], suggesting strong evidence of a protective effect against premature mortality. For instance, a recent meta-analysis [4] based largely on epidemiological studies consisting of large cohorts demonstrated that the risks of more than 25 chronic diseases and mortality were reduced by 20–30% due to routine PA participation. Consequently, encouraging participation in moderate-to-vigorous aerobic exercise has been a key public health strategy especially in prolonging human life [5]. However, most studies to date have focused on the evaluation of leisure time PA [6,7,8] or physical exercise [9] as the main exposure, which has been the major goal of PA promotion policies. In contrast, few studies have focused on domain-specific activities performed during daily routines [10,11], and the effects of household physical activity (HPA) on mortality have been sparsely reported.

HPA, a subdomain of PA, mostly belongs to light-intensity physical activity (LPA) [12,13] but accounts for a large part of people’s daily energy consumption, especially for women. Prior studies [14,15,16] have highlighted the health benefits of increasing the frequency and duration of LPA, and the 2020 WHO guidelines recommend some PA of any intensity (including light intensity) for those not currently meeting moderate-to-vigorous physical activity (MVPA) recommendations [5]. Some subjective studies reported those with lower levels of MVPA (e.g., older and female adults) predominantly engage in HPA [17,18,19], such as shopping and food preparation activities [12,20]. Another objective study reported more LPA and short-bout (1–9 min) MVPA for women because of lots of HPA [21]. Nevertheless, the importance of HPA has not been given sufficient attention. HPA promotion may be a feasible means to increase the volume of activity because HPA does not require dedicated time commitment or planning as it usually involves incidental daily living and an increase in movement during housework [22]. Hence, it is of public and clinical significance to investigate whether these subpopulations can benefit from HPA.

In recent years, studies on the association between HPA and mortality have sprung up [23,24,25,26], but they have primarily focused on all-cause mortality [24,26], and a paucity of exploration on the relationships between HPA and cause-specific mortality remains. Moreover, the available studies have reported inconsistent results that are overwhelmingly concentrated on the elderly population [25,26]. Notably, the vast majority of the studies failed to examine whether this association was independent of sport and physical exercise (SPE). These study gaps were the motivation for this study. Considering there is a clear and deep relationship between PA or LPA participation and health benefits, one would expect that there is an important role of HPA in decreasing mortality. In the present study, we hypothesized that HPA may have a place in the independent effects on all-cause and cause-specific mortality in a national population-based cohort of Chinese adults.

## 2. Methods and Statistical Analyses

### 2.1. Methods

#### 2.1.1. Study Design and Population

The data in this study were derived from the China Family Panel Studies (CFPS), which is a nationally representative, biennial longitudinal survey of Chinese communities, families, and individuals launched by the Institute of Social Science Survey (ISSS) of Peking University. The baseline survey was conducted from April 2010 to March 2011, involving 25 provinces of which the population covers 94.5% of the total population in China except for Hong Kong, Macao, and Taiwan and interviewing 14,960 households and 42,590 individuals. All family members identified in the baseline survey and their future biological/adopted children are defined as CFPS gene members, subject to permanent follow-up of the CFPS survey, and are interviewed every two years. More design and sampling details have been reported previously [27]. The investigation obtained ethical approval from the Peking University Biomedical Ethics Review Committee (approval number: IRB00001052-14010), and all interviewees gave informed consent. 

For this study, 2010 baseline data and four waves of follow-up data (2012, 2014, 2016, and 2018) were extracted from the CFPS to construct a cohort with the view of investigating the relationships of HPA and SPE with all-cause mortality, and mortality from any cardiovascular disease (CVD), respiratory disease (RESP), and cancer. The exclusion criteria were as follows: (1) lost during the succedent follow-up (n = 2549); (2) mortality information was missing from the follow-up survey (n = 191); (3) HPA, SPE information, and important covariates were missing (n = 69). Finally, a cohort of 30,791 subjects followed up from 2010 through 2018 were included in our study. The provincial distribution of adult samples at baseline is presented in Figure 1.

#### 2.1.2. HPA and SPE Assessments

HPA and SPE were assessed by in-person interviews. The self-reported time spent on HPA was ascertained based on the following question item: “In the last month that was not a vacation, how many hours per day on average did you spend participating in the following activities?” This refers to any unpaid labor for the final consumption of one’s family or oneself, such as preparing food, house cleaning, putting clothes and other items in order, and shopping. We examined Cronbach’s alpha (α) to determine the internal consistency reliability of this question to assess whether respondents answered the item in a similar way. The result was confirmed to be above the criterion of at least 0.70 (α = 0.792; 95% CI, 0.788–0.796). Based on the dose–response (D-R) relationships of HPA with all-cause and cause-specific mortality and the summary distribution of our survey data, HPA was classified into four categories: 0 h (non-participant), 0–1 h, 1–3 h, and ≥3 h per day. In addition, the time participating in SPE was also ascertained. Referring to the 2020 WHO guidelines, SPE in our analysis was classified as 0 min/day, 1–60 min/day, and ≥60 min/day considering the summary distribution of our survey data at the same time. The HPA and SPE assessments by intensity are presented in Table 1.

#### 2.1.3. Mortality Outcomes

The endpoints used in this study were all-cause mortality and mortality from CVD, RESP, or cancer. During the follow-up investigation from 2012 to the end of follow-up in CFPS 2018, the survival outcome of participants, including death status, date, and cause of death, were ascertained through CFPS interviews conducted with family members. All-cause mortality was defined as death due to any cause during follow-up, CVD mortality refers to the deceased due to a variety of major circulatory diseases (primarily including acute myocardial infarction, coronary heart disease, stroke), RESP mortality was considered to be death from a variety of respiratory diseases (primarily including pneumonia, emphysema, and pulmonary heart disease), and cancer mortality was identified as death caused by various benign tumor and malignancies.

#### 2.1.4. Covariates

Information about demographic and socioeconomic factors, lifestyle behaviors, and health status was collected by uniformly trained investigators with standardized questionnaires. Demographic and socio-economic status included gender (male and female), age, ethnicity (Han and minority), marital status, residential region (urban and rural areas), geolocation (North and South), education attainment (illiteracy, 1–9 years and >9 years), employment status (current, former, and never), and annual household income (RMB) (low: 0–15,000, medium: 15,000–40,000, and high: ≥40,000). Behavioral factors included smoking status (current, former, and never), alcohol consumption (current, former, and never), nighttime sleep (<6 h/night, 6–8 h/night, ≥8 h/night), SPE (0 min/day, 1–60 min/day, ≥60 min/day). Health status comprised self-reported chronic disease (yes or no) and body mass index (BMI) (underweight: <18.5 kg/m^2^, normal: 18.5–23.9 kg/m^2^, overweight: ≥24.0 kg/m^2^). BMI was calculated as weight in kilograms divided by height in meters squared. The Northern and Southern regions were divided by geographical boundaries comprising the Qinling Mountains and the Huai River. Current and former smokers were regarded as participants who smoked in the last month and a month ago, respectively. Current drinkers referred to drinking at least 3 times/week during the last month and consuming at least 220 milliliters of alcohol each time. Former drinkers were defined as drinking at least 3 times/week a month ago and consuming at least 220 milliliters of alcohol each time (http://www.isss.pku.edu.cn/cfps/en/documentation/questionnaires/index.htm, accessed on 31 December 2022).

### 2.2. Statistical Analyses

Participants were classified into four groups according to the time engaged in HPA: 0 h/day, 0–1 h/day, 1–3 h/day, and ≥3 h/day. Baseline characteristics of participants were described by calculating the mean± standard deviation (SD) for continuous variables (e.g., age and BMI) and percentages for categorical covariates (e.g., gender, ethnicity, and education attainment). 

The associations of HPA and SPE with all-cause mortality and mortality from CVD, RESP, and cancer were summarized using Cox proportional hazard models. Hazard ratios (HRs) and their 95% confidence intervals (CIs) were estimated via both sex- and age-adjusted and multivariable-adjusted models, using the lowest group (0 h/day for HPA and 0 min/day for SPE) as the reference. The multivariable-adjusted models were adjusted for demographic and socioeconomic factors (sex, age, ethnicity, marital status, residential region, geolocation, education attainment, employment status, household income), lifestyle behaviors (smoking status, alcohol consumption, sleep duration), and health status (chronic disease, BMI).

To smooth the D-R relationships of HPA and SPE with all-cause and cause-specific mortality, restricted cubic spline across the PA levels was performed by entering the categorical activity variable as a continuous variable into the Cox model. To test potential effect modifiers in HPA–mortality associations, we performed subgroup analyses by sex. To minimize potential reverse causality and season influence, we conducted sensitivity analyses by excluding (a) individuals with chronic diseases at baseline; (b) confirmed deaths during the first year of follow-up, and by including the interviewing time as a covariate. 

We divided HPA into three groups (0/0–1/≥1 h) and SPE into three levels (0/1–60/≥60 min) to explore the joint effects of HPA and SPE against all-cause and cause-specific mortality (HPA 0 h and SPE 0 min as reference). To investigate the interactive effects of HPA and SPE, we calculated the relative excess risk due to interaction (RERI) and attributable proportion (AP) of additive effects to the total observed effects. We classified the HPA into two groups (i.e., <1 h vs. ≥1 h) using the median as the cut-off point, and categorized SPE as 0, 1–60, and ≥60 min, with groups < 1 h and 1–60 min as the reference.

All statistical analyses were performed with R version 4.1.2 (R Foundation for Statistical Computing, Vienna, Austria). We used the “coxph” function to fit the Cox model, “rms” for smoothing nonlinear terms, and “epiR” for interactive analysis. A two-sided *p* value of <0.05 was regarded as statistically significant. 

## 3. Results

Table 2 describes the baseline characteristics of the study population according to HPA levels. The study involved 30,791 subjects aged 16–110 years with a mean (SD) age of 45.6 (16.3) years, among which 48.3% were men and 91.7% were of Han ethnicity. During a median follow-up of 7.2 years, 1649 cases of all-cause mortality were identified, including 209 from CVD, 123 from RESP, and 323 from cancer. Additionally, 9060 (29.4%), 10,628 (34.5%), and 3775 (12.3%) subjects spent 0–1, 1–3, and ≥3 h participating in HPA per day, respectively. Even though HPA is less intensive and active, it accounts for a large proportion of daily time for most people. Only about one-fourth of the participants did not perform any HPA, and among them, men accounted for over 75%. In contrast, despite being more active and aerobic, over 70% of the population did not do any SPE, while less than 5% of the subjects reported ≥60 min of SPE per day. More than half of the participants lived in the rural community (56.3%) and were from the Northern regions (55.7%). 

Table 3 outlines the associations of HPA and SPE with all-cause and cause-specific mortality. Overall, comparable findings were obtained in both the sex- and age-adjusted and multivariable-adjusted models. For SPE, in the multivariable-adjusted analysis, a significant inverse association was identified for all-cause mortality only, exhibiting a 43% lower risk (HR, 0.57; 95% CI, 0.42–0.78) for the group of ≥60 min/day compared to the inactive reference group of 0 min/day. Meanwhile, largely consistent associations were identified between the HPA groups and mortality outcomes, suggesting remarkably reduced risks of 43–60% in all-cause mortality, 42–50% in CVD mortality, 36–71% in RESP mortality, and 38–46% in cancer mortality. In general, higher levels of HPA tended to be associated with lower risks. For example, the risk of all-cause mortality was reduced to 0.57 (0.50–0.66) in the group of 0–1 h/day of HPA and additionally reduced to 0.50 (0.44–0.58) and 0.40 (0.33–0.48) in the groups of 1–3 and ≥3 h/day of HPA, respectively, compared to the reference. Sensitivity analyses by excluding (a) individuals with chronic diseases at baseline are presented in Appendix A; (b) confirmed deaths during the first year of follow-up in Appendix A; including the interviewing time as a covariate in Appendix A. All of the main results did not materially change. 

Figure 2 shows the D-R relationships of HPA and SPE with all-cause and cause-specific mortality using a restricted cubic spline smoother. For HPA, an approximately inverse J-shape association was identified for all-cause (*P*_nonlinear_ < 0.001), CVD (0.003), RESP (0.082), and cancer (0.026) mortality, suggesting strong evidence of potential nonlinearity. The risks of mortality decreased sharply as the participation time of HPA increased (<2 h/day), while additional beneficial effects tended to be subtle afterwards (>3 h/day). In the case of SPE, we failed to identify significant violations of linear associations for all-cause, CVD, RESP, and cancer, with all *p*-values for nonlinearity >0.2.

Figure 3 summarizes the sex-specific estimates for the associations of HPA and SPE with all-cause and cause-specific mortality. Summarily, compared to men, women performing HPA showed stronger inverse associations with all-cause, CVD, and cancer mortality but not with RESP mortality. For instance, women performing ≥3 h/day of HPA had a 22% lower all-cause mortality (HR, 0.26; 95% CI, 0.20–0.34), compared to men with the same participation time of HPA (HR, 0.48; 95% CI, 0.36–0.64). In terms of SPE, no evident pattern of gender differences was found in the associations with mortality outcomes. For the same contrast of 0 min/day, women had generally lower SPE-associated risk than men in the SPE group of ≥60 min/day for several mortality outcomes (e.g., all-cause mortality), while mixed results were observed in the SPE group of ≥60 min/day for RESP and cancer mortality.

Figure 4 reveals the joint and interactive effects of HPA and SPE against all-cause and cause-specific mortality. Broadly, HPA reduced the risk of death regardless of the augmentation of SPE compared to 0 hour/day of HPA and 0 min/day of SPE, suggesting the independent protective effects of HPA. For instance, spending ≥1 h/day doing HPA was associated with a lessened all-cause mortality risk of 54% (HR, 0.46; 95% CI, 0.37–0.56) for subjects doing SPE 1–60 min/day and 77% (HR, 0.23; 95% CI, 0.13–0.39) for those doing SPE ≥60 min/day. We did not identify significant evidence of additive interactions between HPA and SPE.

## 4. Discussion

To the best of our knowledge, this is the first nationwide prospective cohort study to date to evaluate the association of HPA with cause-specific mortality in China. HPA was found to be significantly associated with lower risks of all-cause and cause-specific mortality, and these associations were independent of SPE-associated effects. In contrast to men with the same participation level of HPA, women exhibited greater risk reduction of all-cause mortality and mortality from CVD and cancer. As the representative of MVPA, SPE did not show such a strong association with death outcomes as expected. In contrast, significant risk reductions were only observed for all-cause mortality, and the risk decreased only for women who participated for ≥60 min/day compared to non-participants in the sex subgroup analysis. Our study may contribute to the existing literature and add significant evidence of the beneficial effects of HPA in reducing all-cause mortality and mortality from any CVD, RESP, and cancer.

Despite the considerable heterogeneity in the effect magnitudes, prior HPA–mortality studies have broadly reported consistent results, declaring a protective effect for mortality in both elder and younger adults from various countries [28,29,30]. Compared to those not involved in HPA, substantial reductions in mortality risk (50–70%) were reported in a US cohort of 948 subjects aged ≥75 years [29] and a representative Spanish cohort consisting of 4008 participants aged ≥65 years [30]. HPA–mortality studies conducted in younger populations have reported similar associations with lower magnitudes of risk reduction [25,28]. For example, another larger (38,379 participants) and younger (30–65 years) Spanish cohort demonstrated a 28–63% lower risk for overall and cause-specific mortalities associated with HPA [28]. Likewise, a German prospective cohort of 4672 men and women, aged 25–74 years, reported 18% and 28% lower risk of all-cause and CVD mortality, respectively [25]. With regard to our cohort sample (ages 16–110 years), slightly greater risk reductions of 36–71% for four mortality categories were identified to be related to HPA. Several plausible mechanisms could account for the observed declines in mortality associated with HPA: (1) LPA appeared to be associated with reduced adiposity, improved blood pressure, and lipidaemia, which could be relevant to obtain health benefits [18]; (2) acting as supplements of total energy expenditure, HPA may reduce insulin resistance and lower fasting insulin levels, decrease the production of inflammatory markers, enhance innate and acquired immune response, and regulate sex hormones, which are supposed to be the protective factors against tumorigenesis [31], angiocardiopathy [32], and diabetes [33]. 

Previous reviews and meta-analyses [34,35,36] have shown an inverse linear association between total PA or MVPA levels and lower mortality. Nevertheless, investigations into the associations between HPA and mortality have seldom been performed in a D-R manner. As far as we can see, ours is the first study to explore HPA–mortality relationships in a D-R pattern and confirm an inverted J-shaped association with a considerable risk reduction of 60% at about 3 h/day. Surprisingly, no further reductions occurred with higher levels than 3 h/day of HPA except for all-cause and RESP mortality, where risk appeared to slightly decrease further. Generally consistent evidence for nonlinear D-R relationships has also been observed in two LPA–mortality studies [37,38]. For instance, a 42% lower mortality risk was associated with 4–5 h/day of LPA in an inverse nonlinear manner in a US prospective study of 4840 adults [37]. Similarly, an approximate inverted J-shaped curve was achieved for associations between LPA and mortality in a new meta-analysis composed of eight LPA–mortality cohorts [38], with maximal risk reductions of 52% at about 6 h/day. Collectively, these studies provide accordant evidence that participating in HPA or LPA may exhibit a similar pattern of nonlinear relationships with mortality, identifying a monotonic risk reduction of 42–60% at 3–6 h/day of HPA or LPA and flat hazard ratios thereafter, which may be powerful evidence for PA recommendations [37]. Owing to the diversity of exposure categories (e.g., MET-h/week [34], duration of PA [37,38], and frequency of PA [35]) adopted in prior investigations, further studies on HPA–mortality associations in a D-R manner are warranted to validate this finding in a broader population across the globe.

Controversial results of the association between HPA and mortality by sex have been previously reported in cohorts with different follow-up periods and among populations of different races and ages [18,28,29,39]. The European Cancer and Nutrition Prospective Cohort—Spanish Study (EPIC-Spain), with a mean follow-up of 13.6 (±1.4) years [28], demonstrated that HPA significantly reduced the risk of all-cause mortality (HR = 0.47, 95% CI: 0.34–0.64) in women, but for men, no association was observed. Completely antipodal results were reported in a Chinese cohort study of 2,867 subjects aged ≥65 years with a 7-year follow-up period, declaring a significant decrease (29–44%) in all-cause mortality among men only [39]. Unlike the above-mentioned HPA–mortality cohorts identifying significant associations in a single sex, several studies have observed risk reduction in both sexes, with greater effect magnitudes either for men [18] or for women [26,29]. For example, a middle-aged (40 to 69 years) Korean cohort with an 8-year follow-up declared that the participation of HPA but not LTPA decreased mortality risk by 28% for men and 16% for women [18], while an older US cohort with a shorter follow-up period (3 years) [29] reported lower mortality risk by 66% for women and 53% for men. Our study endorses a greater benefit in reducing mortality risk among women. For instance, engaging in HPA for ≥3 h/day was associated with a lower risk of all-cause mortality by 72% and 54% for women and men, respectively. Differences in the study design (e.g., follow-up period) and population characteristics (e.g., race, regionalism, and age of participants) [40] could partly account for the sex discrepancy in HPA–mortality associations between studies. 

Some limitations of this study must be acknowledged. First, the HPA and SPE assessments were based on a self-management questionnaire in the CFPS studies, where the investigators only collected information on HPA and SPE over the last month to avoid irresistible recall bias. However, this information could also be influenced by seasonality [41]. Second, since HPA is characterized by less active intensity and substantial variations mainly involving upper body movement, we could have poorly assessed HPA by objective measurements of different types in large population-based prospective studies [14,24,42]. Generally, the self-management questionnaire is considered the most common means for assessing PA and HPA, which has been widely reported in previous relevant studies [4,22]. Third, mortality outcomes were ascertained by interviewing the family members of the deceased in each wave of the survey, resulting in a fraction (3.2%) of undocumented deaths, which may have led, to some extent, to misclassification for cause-specific mortality in our analysis. Fourth, through a variety of potential confounding effects on demographic and socio-economic status, behavioral factors and health status were controlled for, and residual confounding of sedentary time, which is important as the main exposure or an adjusting factor in PA studies, may still exist. Finally, HPA and SPE levels were found to remain fairly stable during the survey period for most of the participants, so we assumed them to be unchanged throughout the follow-up period as reported at baseline. This is a general means in prior epidemiological studies, which was also acknowledged in their limitations [30,43]. Further research is needed to take into account subsequent modifications in PA patterns over a lifetime.

## 5. Conclusions

In summary, our study contributes prospective data that endorse the beneficial effects of HPA on reducing risks of all-cause and cause-specific mortality in Chinese adults. In this study, substantially lower risks of all-cause, CVD, RESP, and cancer mortality were found to be favorably associated with participation in HPA. Generally, these associations were stronger in females. Our findings support previous evidence on the health effect of increasing daily levels of HPA and may provide scientific guidance for making public health policy in the prevention and management of premature mortality. In the future, more studies are warranted worldwide to focus on the long-term effects of HPA on human health.

## Figures and Tables

**Figure 1 ijerph-20-00987-f001:**
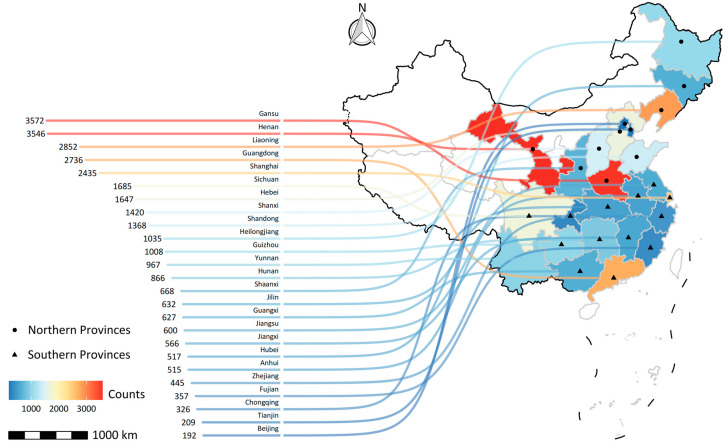
Provincial distribution of adult samples at baseline.

**Figure 2 ijerph-20-00987-f002:**
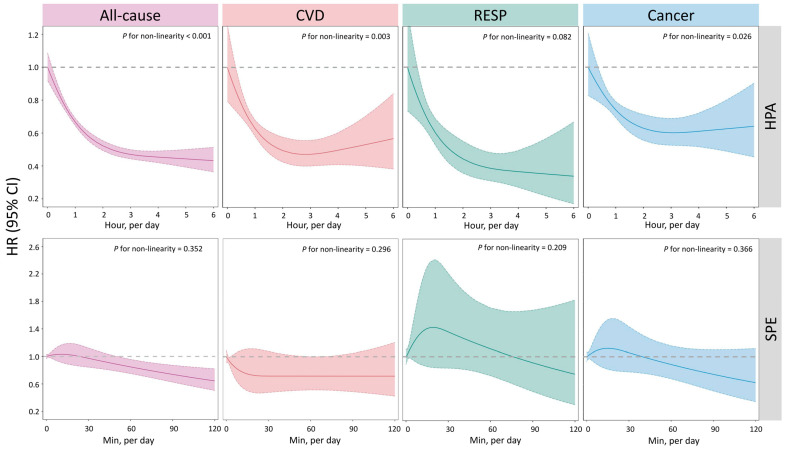
Multivariable-adjusted spline curve for a relation between estimated HPA and SPE with all-cause, CVD, RESP, and cancer mortality. We adjusted for gender, age, BMI, ethnicity, marital status, residential region, geolocation, education attainment, employment status, household income, smoking status, alcohol consumption, sleep duration, and chronic disease. Abbreviations: HPA, household physical activity; SPE, sport and physical exercise; CVD, cardiovascular disease; RESP, respiratory disease; HR, hazard ratio; CI, confidence interval.

**Figure 3 ijerph-20-00987-f003:**
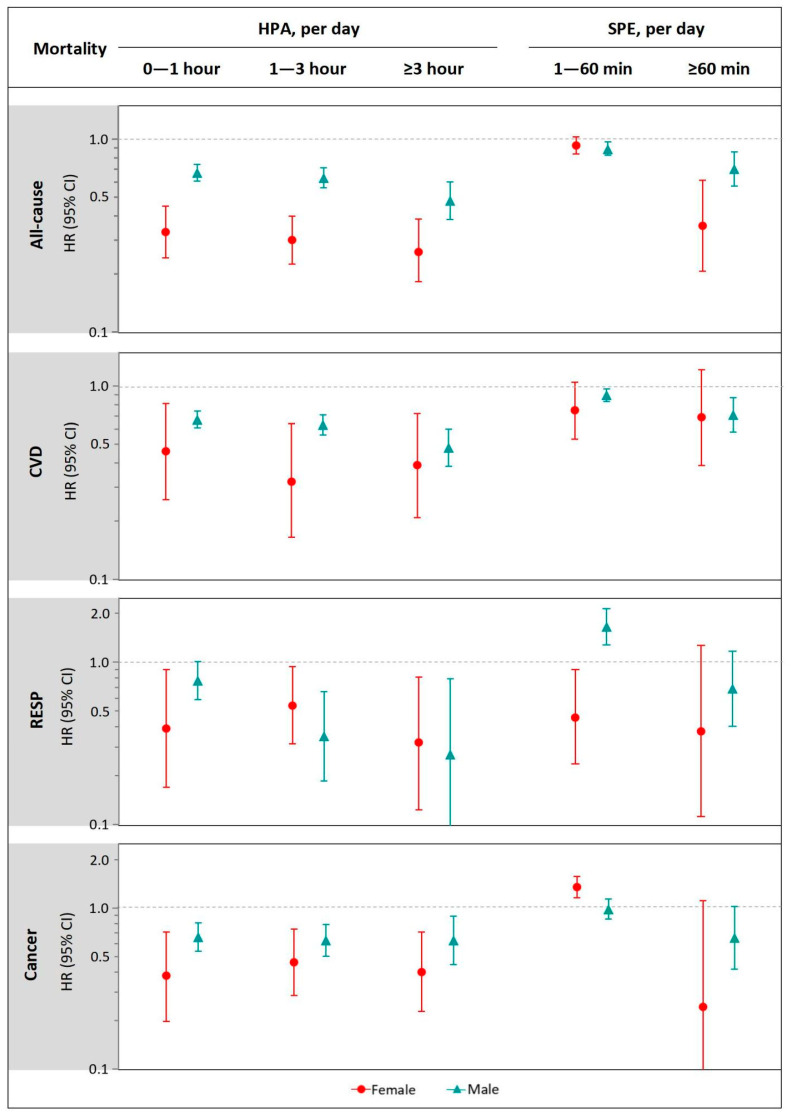
Sex subgroup analysis for association of HPA and SPE with all-cause, CVD, RESP, and cancer mortality. We adjusted for gender, age, BMI, ethnicity, marital status, residential region, geolocation, education attainment, employment status, household income, smoking status, alcohol consumption, sleep duration, and chronic disease. Abbreviations: HPA, household physical activity; SPE, sport and physical exercise; CVD, cardiovascular disease; RESP, respiratory disease; BMI, body mass index; HR, hazard ratio; CI, confidence interval.

**Figure 4 ijerph-20-00987-f004:**
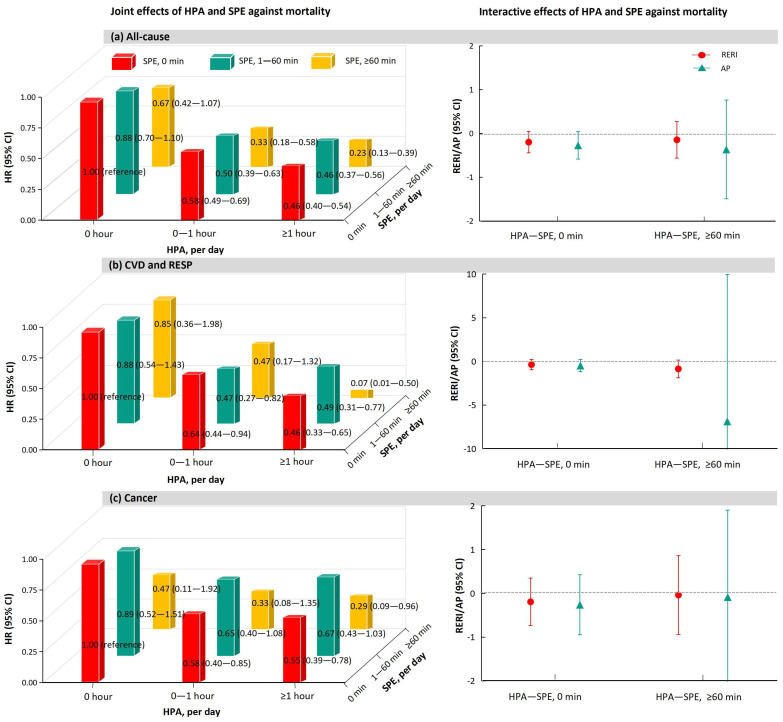
The joint and interactive effects between HPA and SPE against all-cause, CVD, RESP, and cancer mortality. We adjusted for gender, age, BMI, ethnicity, marital status, residential region, geolocation, education attainment, employment status, household income, smoking status, alcohol consumption, sleep duration, and chronic disease. Abbreviations: HPA, household physical activity; SPE, sport and physical exercise; CVD, cardiovascular disease; RESP, respiratory disease; BMI, body mass index; HR, hazard ratio; CI, confidence interval; RERI, relative excess risk due to interaction; AP, attributable proportion. Notes: We did not estimate joint and interactive effects for CVD and RESP mortality, respectively, because the number of RESP deaths for the group of HPA ≥1 h and SPE ≥60 min stratified by cross-classification was zero, which was insufficient to yield results. Therefore, we combined CVD with RESP to perform joint and interactive effect analysis.

**Table 1 ijerph-20-00987-t001:** HPA and SPE assessments by intensity.

Variables	LPA	MVPA
HPA	Preparing food, cleaning afterwards, washing clothes and putting them in order, shopping, feeding pets, arranging and managing family affairs (planning a party, decorating the room, planning a trip, making a shopping list, and searching investment information)	Cleaning up the house and the surroundings, house decorating, maintenance, and do-it-yourself repairs
SPE	Walking, jogging, practicing Tai Chi	Mountain climbing, long-distance running, practicing kinds of Kung Fu, dancing, aerobics, yoga, playing ball games, swimming, diving, rowing, sailing and other kinds of watersports, winter sports, wrestling, judo, boxing, and other kinds of sports with physical contact

Abbreviations: HPA, household physical activity; SPE, sport and physical exercise; LPA, light-intensity physical activity; MVPA, moderate-to-vigorous physical activity.

**Table 2 ijerph-20-00987-t002:** Baseline characteristics of included participants (n = 30,791) by HPA.

Variables	HPA per Day				
	Total	0 h	0–1 h	1–3 h	≥3 h
Population					
Persons, n	30,791	7328	9060	10,628	3775
All-cause death, n	1649	564	383	512	190
CVD	209	72	44	60	33
RESP	123	43	33	34	13
Cancer	323	97	75	112	39
Demographic characteristics					
Age, years	45.6 ± 16.3	41.2 ± 18.3	42.9 ± 15.9	48.6 ± 14.5	52.5 ± 13.6
Male sex, %	48.6	77.7	58.0	30.8	19.4
Han ethnicity, %	91.7	93.4	92.3	90.3	90.6
Married, %	80.7	68.0	79.7	87.4	88.8
Urban, %	43.7	45.4	48.0	40.9	38.2
North, %	55.7	58.0	55.4	53.7	57.4
Education attainment, %					
Illiteracy	24.1	15.4	17.6	30.7	38.0
1–9 years	53.4	51.9	53.7	54.8	51.4
>9 years	22.5	32.6	28.7	14.5	10.6
Annual household income, %					
Low	28.7	23.9	27.1	31.4	33.8
Medium	43.0	44.2	42.5	43.4	40.4
High	28.4	31.9	30.4	25.2	25.8
Employment status, %					
Current	50.6	54.1	59.5	49.4	25.3
Former	29.1	22.6	23.9	32.4	45.4
Never	20.3	23.3	16.6	18.3	29.3
Behavioral factors					
Smoking status, %					
Current	30.4	45.3	35.5	21.2	15.2
Former	6.3	8.6	7.4	4.7	3.5
Never	63.3	46.1	57.1	74.1	81.3
Alcohol consumption, %					
Current	16.1	22.3	19.4	11.8	8.1
Former	4.6	6.3	4.8	3.8	3.4
Never	79.3	71.4	75.9	84.4	88.5
Nighttime sleep, %					
<6 h/night	7.8	8.2	7.1	7.6	9.7
6–8 h/night	63.3	61.3	64.4	63.7	63.2
≥8 h/night	28.9	30.5	28.5	28.7	27.1
SPE, %					
0 min/day	73.2	72.1	68.9	76.1	77.9
1–60 min/day	23.2	24.2	27.3	20.5	19.0
≥60 min/day	3.5	3.7	3.8	3.4	3.1
Health status					
Chronic disease, %					
Yes	14.7	12.0	12.9	16.3	20.2
No	85.3	88.0	87.1	83.7	79.8
BMI, kg/m^2^					
Underweight	10.9	12.4	10.7	9.7	11.7
Normal	62.0	60.6	62.5	63.2	60.3
Overweight	27.1	27.0	26.8	27.1	28.0

Notes: Data are presented using mean ± SD for continuous variables and percentages for categorical variables. Abbreviations: HPA, household physical activity; SPE, sport and physical exercise; CVD, cardiovascular disease; RESP, respiratory disease; BMI, body mass index.

**Table 3 ijerph-20-00987-t003:** Effects of HPA and SPE on all-cause, CVD, RESP, and cancer mortality.

Mortality Outcome	HPA per Day				SPE per Day		
	0 h	0–1 h	1–3 h	≥3 h	0 min	1–60 min	≥60 min
**All-cause**							
Sex- and age-adjusted	Ref.	0.52	0.46	0.39	Ref.	0.78	0.58
model		(0.45–0.59)	(0.41–0.53)	(0.32–0.46)		(0.69–0.88)	(0.44–0.76)
Multivariable-adjusted	Ref.	0.57	0.50	0.40	Ref.	0.91	0.57
model ^a^		(0.50–0.66)	(0.44–0.58)	(0.33–0.48)		(0.79–1.04)	(0.42–0.78)
**CVD**							
Sex- and age-adjusted	Ref.	0.47	0.44	0.55	Ref.	0.85	0.64
model		(0.33–0.69)	(0.31–0.64)	(0.36–0.86)		(0.62–1.78)	(0.32–1.31)
Multivariable-adjusted	Ref.	0.58	0.50	0.51	Ref.	0.79	0.58
model ^a^		(0.38–0.87)	(0.34–0.74)	(0.31–0.84)		(0.54–1.15)	(0.27–1.22)
**RESP**							
Sex- and age-adjusted	Ref.	0.58	0.39	0.32	Ref.	0.91	0.40
model		(0.37–0.92)	(0.24–0.63)	(0.17–0.62)		(0.61–1.37)	(0.13–1.25)
Multivariable-adjusted	Ref.	0.64	0.44	0.29	Ref.	1.16	0.54
model ^a^		(0.39–1.07)	(0.26–0.74)	(0.14–0.63)		(0.72–1.89)	(0.16–1.77)
**Cancer**							
Sex- and age-adjusted	Ref.	0.61	0.69	0.60	Ref.	1.00	0.44
model		(0.33–0.69)	(0.31–0.64)	(0.36–0.86)		(0.77–1.28)	(0.21–0.94)
Multivariable-adjusted	Ref.	0.62	0.62	0.54	Ref.	1.08	0.51
model ^a^		(0.45–0.85)	(0.45–0.84)	(0.35–0.83)		(0.81–1.45)	(0.24–1.11)

^a^ We adjusted for gender, age, BMI, ethnicity, marital status, residential region, geolocation, education attainment, employment status, household income, smoking status, alcohol consumption, sleep duration, and chronic disease. Abbreviations: HPA, household physical activity; SPE, sport and physical exercise; CVD, cardiovascular disease; RESP, respiratory disease; BMI, body mass index; HR, hazard ratio; CI, confidence interval.

## Data Availability

The data that support the findings of this study are available from the CFPS project site (https://www.isss.pku.edu.cn/cfps/en/ (accessed on 31 December 2022)).

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
