# Peer review of "Participation in Household Physical Activity Lowers Mortality Risk in Chinese Women and Men"

_ijerph, 2023, doi:10.3390/ijerph20020987_

Round 1

Reviewer 1 Report

The manuscript presented to me for review is a large-scale study (over 30,000 participants) of the impact of household physical activity on mortality in a cohort of Chinese adults conducted over a long period (since 2010). The results confirm that this type of physical activity lowers the risk of all-cause and cause specific mortality in this population.

The introduction is well written, the study methods and statistical analysis are described correctly and do not raise my objections. The results are compiled and presented in a very transparent manner. The tables added as supplementary materials enhance the quality of presentation of the results. The discussion is well conducted. An important element are the added study limitations - with which I agree. My only suggestion is to add at the end of Introduction study hypothesis and its rationale based on literature cited in this chapter.

I suggest to accept this manuscript for publication after minor corrections.

Reviewer 2 Report

The manuscript provides the findings of a study conducted with objective to investigate the independent effects of household physical activity on all-cause, cardiovascular disease, respiratory disease and cancer mortality in a national population-based cohort of Chinese adults. Interesting study and there is a component of originality in the research topic. The theoretical basis is well founded and justifies the study. The bibliographic references are current and relevant to the subject of study. The methods used are sufficiently described, appropriate and consistent with the study proposal. The results are presented in an attractive manner, and the discussion of the findings show good prospects for the area of public health. The potential limitations are acknowledged. The conclusions are consistent with presented evidence and arguments. The findings of the study have good prospects to offer important contributions to the knowledge area, and offers important support for the design and conduct of future studies. However, since an immediate question was used to measure the self-reported time spending on household physical activity of study participants, I suggest that the authors add in the methods section information about the psychometric properties of this question.

Reviewer 3 Report

Dear Authors,

Thank you for an interesting paper.

I read your paper and came up with some ideas

Please check my comments.

I hope my comments will improve your manuscript.

Best,

Round 2

Reviewer 3 Report

Dear Authors,

Thank you for an interesting paper.

I checked the revised version manuscript.

Currently, you politely replied and properly modified it.  

Comprehensively, I feel yours is well documented.

Your manuscript becomes interesting and beneficial for readers.

Best,